# MATRIX CAPSULES WITH EM ROUTING

**Geoffrey Hinton, Sara Sabour, Nicholas Frosst**
Google Brain
Toronto, Canada
`{geoffhinton, sasabour, frosst}@google.com`

## ABSTRACT

A capsule is a group of neurons whose outputs represent different properties of the same entity. Each layer in a capsule network contains many capsules. We describe a version of capsules in which each capsule has a logistic unit to represent the presence of an entity and a 4x4 matrix which could learn to represent the relationship between that entity and the viewer (the pose). A capsule in one layer votes for the pose matrix of many different capsules in the layer above by multiplying its own pose matrix by trainable viewpoint-invariant transformation matrices that could learn to represent part-whole relationships. Each of these votes is weighted by an assignment coefficient. These coefficients are iteratively updated for each image using the Expectation-Maximization algorithm such that the output of each capsule is routed to a capsule in the layer above that receives a cluster of similar votes. The transformation matrices are trained discriminatively by backpropagating through the unrolled iterations of EM between each pair of adjacent capsule layers. On the smallNORB benchmark, capsules reduce the number of test errors by 45% compared to the state-of-the-art. Capsules also show far more resistance to white box adversarial attacks than our baseline convolutional neural network.

## 1 INTRODUCTION

Convolutional neural nets are based on the simple fact that a vision system needs to use the same knowledge at all locations in the image. This is achieved by tying the weights of feature detectors so that features learned at one location are available at other locations. Convolutional capsules extend the sharing of knowledge across locations to include knowledge about the part-whole relationships that characterize a familiar shape. Viewpoint changes have complicated effects on pixel intensities but simple, linear effects on the pose matrix that represents the relationship between an object or object-part and the viewer. The aim of capsules is to make good use of this underlying linearity, both for dealing with viewpoint variations and for improving segmentation decisions.

Capsules use high-dimensional coincidence filtering: a familiar object can be detected by looking for agreement between votes for its pose matrix. These votes come from parts that have already been detected. A part produces a vote by multiplying its own pose matrix by a learned transformation matrix that represents the viewpoint invariant relationship between the part and the whole. As the viewpoint changes, the pose matrices of the parts and the whole will change in a coordinated way so that any agreement between votes from different parts will persist.

Finding tight clusters of high-dimensional votes that agree in a mist of irrelevant votes is one way of solving the problem of assigning parts to wholes. This is non-trivial because we cannot grid the high-dimensional pose space in the way the low-dimensional translation space is gridded to facilitate convolutions. To solve this challenge, we use a fast iterative process called "routing-by-agreement" that updates the probability with which a part is assigned to a whole based on the proximity of the vote coming from that part to the votes coming from other parts that are assigned to that whole. This is a powerful segmentation principle that allows knowledge of familiar shapes to derive segmentation, rather than just using low-level cues such as proximity or agreement in color or velocity. An important difference between capsules and standard neural nets is that the activation of a capsule is based on a comparison between multiple incoming pose predictions whereas in a standard neural net it is based on a comparison between a single incoming activity vector and a learned weight vector.

## 2 How capsules work

Neural nets typically use simple non-linearities in which a non-linear function is applied to the scalar output of a linear filter. They may also use softmax non-linearities that convert a whole vector of logits into a vector of probabilities. Capsules use a much more complicated non-linearity that converts the whole set of activation probabilities and poses of the capsules in one layer into the activation probabilities and poses of capsules in the next layer.

A capsule network consists of several layers of capsules. The set of capsules in layer $L$ is denoted as $\Omega_L$. Each capsule has a 4x4 pose matrix, $M$, and an activation probability, $a$. These are like the activities in a standard neural net: they depend on the current input and are not stored. In between each capsule $i$ in layer $L$ and each capsule $j$ in layer $L+1$ is a 4x4 trainable transformation matrix, $W_{ij}$. These $W_{ij}$s (and two learned biases per capsule) are the only stored parameters and they are learned discriminatively. The pose matrix of capsule $i$ is transformed by $W_{ij}$ to cast a vote $V_{ij} = M_i W_{ij}$ for the pose matrix of capsule $j$. The poses and activations of all the capsules in layer $L+1$ are calculated by using a non-linear routing procedure which gets as input $V_{ij}$ and $a_i$ for all $i \in \Omega_L, j \in \Omega_{L+1}$.

The non-linear procedure is a version of the Expectation-Maximization procedure. It iteratively adjusts the means, variances, and activation probabilities of the capsules in layer $L+1$ and the assignment probabilities between all $i \in \Omega_L, j \in \Omega_{L+1}$. In appendix 1, we give a gentle intuitive introduction to routing-by-agreement and describe in detail how it relates to the EM algorithm for fitting a mixture of Gaussians.

## 3 Using EM for routing-by-agreement

Let us suppose that we have already decided on the poses and activation probabilities of all the capsules in a layer and we now want to decide which capsules to activate in the layer above and how to assign each active lower-level capsule to one active higher-level capsule. Each capsule in the higher-layer corresponds to a Gaussian and the pose of each active capsule in the lower-layer (converted to a vector) corresponds to a data-point (or a fraction of a data-point if the capsule is partially active).

Using the minimum description length principle we have a choice when deciding whether or not to activate a higher-level capsule. **Choice 0**: if we do not activate it, we must pay a fixed cost of $-\beta_u$ per data-point for describing the poses of all the lower-level capsules that are assigned to the higher-level capsule. This cost is the negative log probability density of the data-point under an improper uniform prior. For fractional assignments we pay that fraction of the fixed cost. **Choice 1**: if we do activate the higher-level capsule we must pay a fixed cost of $-\beta_a$ for coding its mean and variance and the fact that it is active and then pay additional costs, pro-rated by the assignment probabilities, for describing the discrepancies between the lower-level means and the values predicted for them when the mean of the higher-level capsule is used to predict them via the inverse of the transformation matrix. A much simpler way to compute the cost of describing a datapoint is to use the negative log probability density of that datapoint's vote under the Gaussian distribution fitted by whatever higher-level capsule it gets assigned to. This is incorrect for reasons explained in appendix 1, but we use it because it requires much less computation (also explained in the appendix). The difference in cost between choice 0 and choice 1, is then put through the logistic function on each iteration to determine the higher-level capsule's activation probability. Appendix 1 explains why the logistic is the correct function to use.

Using our efficient approximation for choice 1 above, the incremental cost of explaining a whole data-point $i$ by using an active capsule $j$ that has an axis-aligned covariance matrix is simply the sum over all dimensions of the cost of explaining each dimension, $h$, of the vote $V_{ij}$. This is simply $-ln(P_{i|j}^h)$ where $P_{i|j}^h$ is the probability density of the $h^{th}$ component of the vectorized vote $V_{ij}$ under $j$'s Gaussian model for dimension $h$ which has variance $(\sigma_j^h)^2$ and mean $\mu_j^h$ where $\mu_j$ is the vectorized version of $j$'s pose matrix $M_j$.

$$P_{i|j}^h = \frac{1}{\sqrt{2\pi(\sigma_j^h)^2}} \exp\left(-\frac{(V_{ij}^h - \mu_j^h)^2}{2(\sigma_j^h)^2}\right), \qquad ln(P_{i|j}^h) = -\frac{(V_{ij}^h - \mu_j^h)^2}{2(\sigma_j^h)^2} - ln(\sigma_j^h) - ln(2\pi)/2$$

Summing over all lower-level capsules for a single dimension, $h$, of $j$ we get:

$$
\begin{aligned}
cost_j^h &= \sum_i -r_{ij} ln(P_{i|j}^h) \\
&= \frac{\sum_i r_{ij}(V_{ij}^h - \mu_j^h)^2}{2(\sigma_j^h)^2} + \left(ln(\sigma_j^h) + \frac{ln(2\pi)}{2}\right)\sum_i r_{ij} \\
&= \left(ln(\sigma_j^h) + \frac{1}{2} + \frac{ln(2\pi)}{2}\right)\sum_i r_{ij}
\end{aligned}
\tag{1}
$$

where $\sum_i r_{ij}$ is the amount of data assigned to $j$ and $V_{ij}^h$ is the value on dimension $h$ of $V_{ij}$. Turning-on capsule $j$ *increases* the description length for the means of the lower-level capsules assigned to $j$ from $-\beta_u$ per lower-level capsule to $-\beta_a$ plus the sum of the cost over all dimensions so we define the activation function of capsule $j$ to be:

$$
a_j = logistic\left(\lambda\left(\beta_a - \beta_u \sum_i r_{ij} - \sum_h cost_j^h\right)\right)
\tag{2}
$$

where $\beta_a$ is the same for all capsules and $\lambda$ is an inverse temperature parameter. We learn $\beta_a$ and $\beta_u$ discriminatively and set a fixed schedule for $\lambda$ as a hyper-parameter.

For finalizing the pose parameters and activations of the capsules in layer $L + 1$ we run the EM algorithm for few iterations (normally 3) after the pose parameters and activations have already been finalized in layer $L$. The non-linearity implemented by a whole capsule layer is a form of cluster finding using the EM algorithm, so we call it **EM Routing**.

---

**Procedure 1** Routing algorithm returns **activation** and **pose** of the capsules in layer $L + 1$ given the **activations** and **votes** of capsules in layer $L$. $V_{ij}^h$ is the $h^{th}$ dimension of the vote from capsule $i$ with activation $a_i$ in layer $L$ to capsule $j$ in layer $L + 1$. $\beta_a, \beta_u$ are learned discriminatively and the inverse temperature $\lambda$ increases at each iteration with a fixed schedule.

---

1: **procedure** EM ROUTING($\boldsymbol{a}, V$)
2:    $\forall i \in \Omega_L, j \in \Omega_{L+1}: R_{ij} \leftarrow 1/|\Omega_{L+1}|$
3:    **for** $t$ iterations **do**
4:        $\forall j \in \Omega_{L+1}$: M-STEP($\boldsymbol{a}, R, V, j$)
5:        $\forall i \in \Omega_L$: E-STEP($\mu, \sigma, \boldsymbol{a}, V, i$)
     **return** $\boldsymbol{a}, M$

1: **procedure** M-STEP($\boldsymbol{a}, R, V, j$)                    ▷ for one higher-level capsule, $j$
2:    $\forall i \in \Omega_L: R_{ij} \leftarrow R_{ij} * \boldsymbol{a}_i$
3:    $\forall h: \mu_j^h \leftarrow \frac{\sum_i R_{ij} V_{ij}^h}{\sum_i R_{ij}}$
4:    $\forall h: (\sigma_j^h)^2 \leftarrow \frac{\sum_i R_{ij}(V_{ij}^h - \mu_j^h)^2}{\sum_i R_{ij}}$
5:    $cost^h \leftarrow \left(\beta_u + log(\sigma_j^h)\right)\sum_i R_{ij}$
6:    $a_j \leftarrow logistic(\lambda(\beta_a - \sum_h cost^h))$

1: **procedure** E-STEP($\mu, \sigma, \boldsymbol{a}, V, i$)                    ▷ for one lower-level capsule, $i$
2:    $\forall j \in \Omega_{L+1}: \boldsymbol{p}_j \leftarrow \frac{1}{\sqrt{\prod_h^H 2\pi(\sigma_j^h)^2}} \exp\left(-\sum_h^H \frac{(V_{ij}^h - \boldsymbol{\mu}_j^h)^2}{2(\sigma_j^h)^2}\right)$
3:    $\forall j \in \Omega_{L+1}: \boldsymbol{R}_{ij} \leftarrow \frac{\boldsymbol{a}_j \boldsymbol{p}_j}{\sum_{k \in \Omega_{L+1}} \boldsymbol{a}_k \boldsymbol{p}_k}$

---

## 4 THE CAPSULES ARCHITECTURE

The general architecture of our model is shown in Fig. 1. The model starts with a 5x5 convolutional layer with 32 channels (A=32) and a stride of 2 with a ReLU non-linearity. All the other layers are capsule layers starting with the primary capsule layer. The 4x4 pose of each of the B=32 primary capsule types is a learned linear transformation of the output of all the lower-layer ReLUs centered at

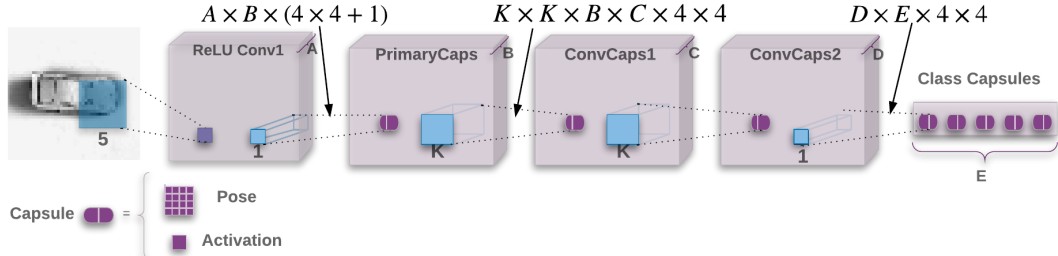

Figure 1: A network with one ReLU convolutional layer followed by a primary convolutional capsule layer and two more convolutional capsule layers.

that location. The activations of the primary capsules are produced by applying the sigmoid function to the weighted sums of the same set of lower-layer ReLUs.

The primary capsules are followed by two 3x3 convolutional capsule layers (K=3), each with 32 capsule types (C=D=32) with strides of 2 and one, respectively. The last layer of convolutional capsules is connected to the final capsule layer which has one capsule per output class.

When connecting the last convolutional capsule layer to the final layer we do not want to throw away information about the location of the convolutional capsules but we also want to make use of the fact that all capsules of the same type are extracting the same entity at different positions. We therefore share the transformation matrices between different positions of the same capsule type and add the scaled coordinate (row, column) of the center of the receptive field of each capsule to the first two elements of the right-hand column of its vote matrix. We refer to this technique as Coordinate Addition. This should encourage the shared final transformations to produce values for those two elements that represent the fine position of the entity relative to the center of the capsule's receptive field.

The routing procedure is used between each adjacent pair of capsule layers. For convolutional capsules, each capsule in layer $L + 1$ sends feedback only to capsules within its receptive field in layer $L$. Therefore each convolutional instance of a capsule in layer $L$ receives at most *kernel_size* X *kernel_size* feedback from each capsule type in layer $L + 1$. The instances closer to the border of the image receive fewer feedbacks with corner ones receiving only one feedback per capsule type in layer $L + 1$.

### 4.1 SPREAD LOSS

In order to make the training less sensitive to the initialization and hyper-parameters of the model, we use "spread loss" to directly maximize the gap between the activation of the target class ($a_t$) and the activation of the other classes. If the activation of a wrong class, $a_i$, is closer than the margin, $m$, to $a_t$ then it is penalized by the squared distance to the margin:

$$L_i = (max(0, m - (a_t - a_i)))^2, \quad L = \sum_{i \neq t} L_i \tag{3}$$

By starting with a small margin of $0.2$ and linearly increasing it during training to $0.9$, we avoid dead capsules in the earlier layers. Spread loss is equivalent to squared Hinge loss with $m = 1$. Guermeur & Monfrini (2011) studies a variant of this loss in the context of multi class SVMs.

## 5 EXPERIMENTS

The smallNORB dataset (LeCun et al. (2004)) has gray-level stereo images of 5 classes of toys: airplanes, cars, trucks, humans and animals. There are 10 physical instances of each class which are painted matte green. 5 physical instances of a class are selected for the training data and the other 5 for the test data. Every individual toy is pictured at 18 different azimuths (0-340), 9 elevations and 6 lighting conditions, so the training and test sets each contain 24,300 stereo pairs of 96x96 images. We selected smallNORB as a benchmark for developing our capsules system because it is carefully designed to be a pure shape recognition task that is not confounded by context and color, but it is much closer to natural images than MNIST.

Table 1: The effect of varying different components of our capsules architecture on smallNORB.

| Routing iterations | Pose structure | Loss | Coordinate Addition | Test error rate |
|---|---|---|---|---|
| 1 | Matrix | Spread | Yes | 9.7% |
| 2 | Matrix | Spread | Yes | 2.2% |
| 3 | Matrix | Spread | Yes | **1.8**% |
| 5 | Matrix | Spread | Yes | 3.9% |
| 3 | Vector | Spread | Yes | 2.9% |
| 3 | Matrix | Spread | No | 2.6% |
| 3 | Vector | Spread | No | 3.2% |
| 3 | Matrix | Margin[1] | Yes | 3.2% |
| 3 | Matrix | CrossEnt | Yes | 5.8% |
| Baseline CNN with 4.2M parameters | | | | 5.2% |
| CNN of Cireşan et al. (2011) with extra input images & deformations | | | | 2.56% |
| Our Best model (third row), with multiple crops during testing | | | | **1.4**% |

We downsample smallNORB to $48 \times 48$ pixels and normalize each image to have zero mean and unit variance. During training, we randomly crop $32 \times 32$ patches and add random brightness and contrast to the cropped images. During test, we crop a $32 \times 32$ patch from the center of the image and achieve $\mathbf{1.8}\%$ test error on smallNORB. If we average the class activations over multiple crops at test time we achieve $1.4\%$. The best reported result on smallNORB without using meta data is $2.56\%$ (Cireşan et al. (2011)). To achieve this, they added two additional stereo pairs of input images that are created by using an on-center off-surround filter and an off-center on-surround filter. They also applied affine distortions to the images. Our work also beats the Sabour et al. (2017) capsule work which achieves $2.7\%$ on smallNORB. We also tested our model on NORB which is a jittered version of smallNORB with added background and we achieved a $2.6\%$ error rate which is on par with the state-of-the-art of $2.7\%$ (Ciresan et al. (2012)).

As the baseline for our experiments on generalization to novel viewpoints we train a CNN which has two convolutional layers with $32$ and $64$ channels respectively. Both layers have a kernel size of $5$ and a stride of $1$ with a $2 \times 2$ max pooling. The third layer is a $1024$ unit fully connected layer with dropout and connects to the 5-way softmax output layer. All hidden units use the ReLU non-linearity. We use the same image preparation for the CNN baseline as described above for the capsule network. Our baseline CNN was the result of an extensive hyperparameter search over filter sizes, numbers of channels and learning rates.

The CNN baseline achieves $5.2\%$ test error rate on smallNORB and has 4.2M parameters. We deduce that the Cireşan et al. (2011) network has 2.7M parameters. By using small matrix multiplies, we reduced the number of parameters by a factor of 15 to 310K compared with our baseline CNN (and a factor of 9 w.r.t Cireşan et al. (2011)). A smaller capsule network of $A = 64, B = 8, C = D = 16$ with only 68K trainable parameters achieves $2.2\%$ test error rate which also beats the prior state-of-the-art.

Fig. 2 shows how EM routing adjusts the vote assignments and the capsule means to find the tight clusters in the votes. The histograms show the distribution of vote distances to the mean (pose) of each class capsule during routing iterations. At the first iteration, votes are distributed equally between 5 final layer capsules. Therefore, all capsules receive votes closer than $0.05$ to their calculated mean. In the second iteration, the assignment probability for agreeing votes increases. Therefore, most of the votes are assigned to the detected clusters, the animal and human class in the middle row, and the other capsules only receive scattered votes which are further than $0.05$ from the calculated mean. The zoomed-out version of Fig. 2 in the Appendix shows the full distribution of vote distances at each routing iteration.

Instead of using our MDL-derived capsule activation term which computes a separate activation probability per capsule, we could view the capsule activations like the mixing proportions in a mixture of Gaussians and set them to be proportional to the sum of the assignment probabilities of a capsule and to sum to $1$ over all the capsules in a layer. This increases the test error rate on

---

[1]The loss proposed by Sabour et al. (2017) for training Capsules.

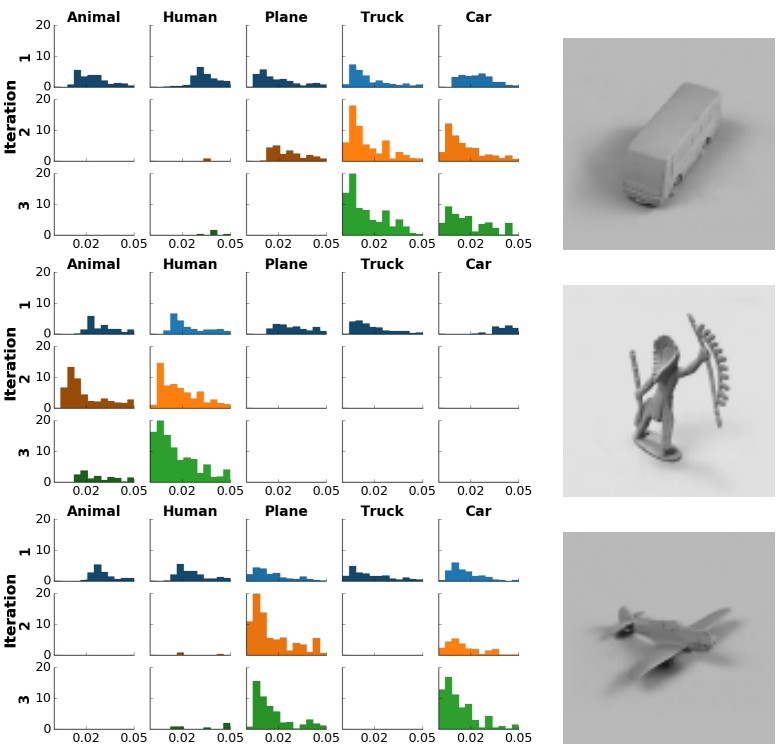

Figure 2: Histogram of distances of votes to the mean of each of the 5 final capsules after each routing iteration. Each distance point is weighted by its assignment probability. All three images are selected from the smallNORB test set. The routing procedure correctly routes the votes in the truck and the human example. The plane example shows a rare failure case of the model where the plane is confused with a car in the third routing iteration. The histograms are zoomed-in to visualize only votes with distances less than 0.05. Fig. B.2 shows the complete histograms for the "human" capsule without clipping the x-axis or fixing the scale of the y-axis.

Table 2: A comparison of the smallNORB test error rate of the baseline CNN and the capsules model on novel viewpoints when both models are matched on error rate for familiar viewpoints.

| Test set | Azimuth | | Elevation | |
|---|---|---|---|---|
| | CNN | Capsules | CNN | Capsules |
| Novel viewpoints | 20% | 13.5% | 17.8% | 12.3% |
| Familiar viewpoints | 3.7% | 3.7% | 4.3% | 4.3% |

smallNORB to $4.5\%$. Tab. 1 summarizes the effects of the number of routing iterations, the type of loss, and the use of matrices rather than vectors for the poses.

The same capsules architecture as Fig. 1 achieves $0.44\%$ test error rate on MNIST. If the number of channels in the first hidden layer is increased to 256, it achieves $11.9\%$ test error rate on Cifar10 (Krizhevsky & Hinton (2009)).

## 5.1 GENERALIZATION TO NOVEL VIEWPOINTS

A more severe test of generalization is to use a limited range of viewpoints for training and to test on a much wider range. We trained both our convolutional baseline and our capsule model on one-third of the training data containing azimuths of (300, 320, 340, 0, 20, 40) and tested on the two-thirds of the test data that contained azimuths from 60 to 280. In a separate experiment, we trained on the 3 smaller elevations and tested on the 6 larger elevations.

It is hard to decide if the capsules model is better at generalizing to novel viewpoints because it achieves better test accuracy on all viewpoints. To eliminate this confounding factor, we stopped training the capsule model when its performance matched the baseline CNN on the third of the test set that used the training viewpoints. Then, we compared these matched models on the two-thirds of the test set with novel viewpoints. Results in Tab. 2 show that compared with the baseline CNN capsules with matched performance on familiar viewpoints reduce the test error rate on novel viewpoints by about $30\%$ for both novel azimuths and novel elevations.

## 6 ADVERSARIAL ROBUSTNESS

There is growing interest in the vulnerability of neural networks to adversarial examples; inputs that have been slightly changed by an attacker to trick a neural net classifier into making the wrong classification. These inputs can be created in a variety of ways, but straightforward strategies such as FGSM (Goodfellow et al. (2014)) have been shown to drastically decrease accuracy in convolutional neural networks on image classification tasks. We compare our capsule model and a traditional convolutional model on their ability to withstand such attacks.

FGSM computes the gradient of the loss w.r.t. each pixel intensity and then changes the pixel intensity by a fixed amount $\epsilon$ in the direction that increases the loss. So the changes only depend on the sign of the gradient at each pixel. This can be extended to a targeted attack by updating the input to maximize the classification probability of a particular wrong class. We generated adversarial attacks using FGSM because it has only one hyper-parameter and it is easy to compare models that have very different gradient magnitudes. To test the robustness of our model, we generated adversarial images from the test set using a fully trained model. We then reported the accuracy of the model on these images.

We found that our model is significantly less vulnerable to both general and targeted FGSM adversarial attacks; a small $\epsilon$ can be used to reduce a convolutional model's accuracy much more than an equivalent $\epsilon$ can on the capsule model (Fig. 3). It should also be noted that the capsule model's accuracy after the untargeted attack never drops below chance ($20\%$) whereas the convolutional model's accuracy is reduced to significantly below chance with an $\epsilon$ as small as $0.2$.

We also tested our model on the slightly more sophisticated adversarial attack of the Basic Iterative Method (Kurakin et al. (2016)), which is simply the aforementioned attack except it takes multiple smaller steps when creating the adversarial image. Here too we find that our model is much more robust to the attack than the traditional convolutional model.

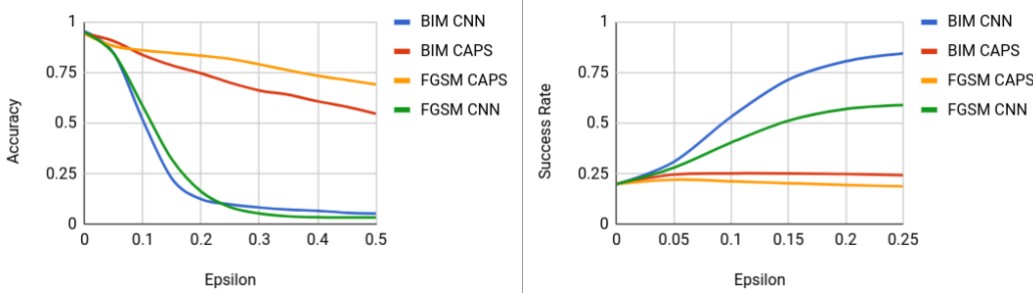

Figure 3: Accuracy against $\epsilon$ after an adversarial attack (left) and Success Rate after a targeted adversarial attack (right). The targeted attack results were evaluated by averaging the success rate after the attack for each of the 5 possible classes.

It has been shown that some robustness to adversarial attacks in models can be due to simple numerical instability in the calculation of the gradient Brendel & Bethge (2017). To ensure that this was not the sole cause of our model's robustness, we calculated the percentage of zero values in the gradient with respect to the image in the capsule model and found it to be smaller than that of the CNN. Furthermore, the capsule gradients, although smaller that those of the CNN, are only smaller by 2 orders of magnitude, as opposed to 16 orders of magnitude seen in Brendel & Bethge (2017)'s work.

Finally we tested our model's robustness to black box attacks by generating adversarial examples with a CNN and testing them on both our capsule model and a different CNN. We found that the capsule model did not perform noticeably better at this task than the CNN.

## 7 RELATED WORK

Among the multiple recent attempts at improving the ability of neural networks to deal with viewpoint variations, there are two main streams. One stream attempts to achieve viewpoint invariance and the other aims for viewpoint equivariance. The work presented by Jaderberg et al. (2015)), Spatial Transformer Networks, seeks viewpoint invariance by changing the sampling of CNNs according to a selection of affine transformations. De Brabandere et al. (2016) extends spatial transformer networks where the filters are adapted during inference depending on the input. They generate different filters for each locality in the feature map rather than applying the same transformation to all filters. Their approach is a step toward input covariance detection from traditional pattern matching frameworks like standard CNNs (LeCun et al. (1990)). Dai et al. (2017) improves upon spatial transformer networks by generalizing the sampling method of filters. Our work differs substantially in that a unit is not activated based on the matching score with a filter (either fixed or dynamically changing during inference). In our case, a capsule is activated only if the transformed poses coming from the layer below match each other. This is a more effective way to capture covariance and leads to models with many fewer parameters that generalize better.

The success of CNNs has motivated many researchers to extend the translational equivariance built in to CNNs to include rotational equivariance (Cohen & Welling (2016), Dieleman et al. (2016), Oyallon & Mallat (2015)). The recent approach in Harmonic Networks (Worrall et al. (2017)) achieves rotation equivariant feature maps by using circular harmonic filters and returning both the maximal response and orientation using complex numbers. This shares the basic representational idea of capsules: By assuming that there is only one instance of the entity at a location, we can use several different numbers to represent its properties. They use a fixed number of streams of rotation orders. By enforcing the equality of the sum of rotation orders along any path, they achieve patch-wise rotation equivariance. This approach is more parameter-efficient than data augmentation approaches, duplicating feature maps, or duplicating filters (Fasel & Gatica-Perez (2006), Laptev et al. (2016)). Our approach encodes general viewpoint equivariance rather than only affine 2D rotations. Symmetry networks (Gens & Domingos (2014)) use iterative Lucas-Kanade optimization to find poses that are supported by the most low-level features. Their key weakness is that the iterative algorithm always starts at the same pose, rather than the mean of the bottom-up votes.

Lenc & Vedaldi (2016) proposes a feature detection mechanism (DetNet) that is equivariant to affine transformations. DetNet is designed to detect the same points in the image under different viewpoint variations. This effort is orthogonal to our work but DetNet might be a good way to implement the de-rendering first-stage that activates the layer of primary capsules.

Our routing algorithm can be seen as an attention mechanism. In this view, it is related to the work of Gregor et al. (2015), where they improved the decoder performance in a generative model by using Gaussian kernels to attend to different parts of the feature map generated by the encoder. Vaswani et al. (2017) uses a softmax attention mechanism to match parts of the query sequence to parts of the input sequence for the translation task and when generating an encoding for the query. They show improvement upon previous translation efforts using recurrent architectures. Our algorithm has attention in the opposite direction. The competition is not between the lower-level capsules that a higher-level capsule might attend to. It is between the higher-level capsules that a lower-level capsule might send its vote to.

### 7.1 PREVIOUS WORK ON CAPSULES

Hinton et al. (2011) used a transformation matrix in a transforming autoencoder that learned to transform a stereo pair of images into a stereo pair from a slightly different viewpoint. However, that system requires the transformation matrix to be supplied externally. More recently, routing-by-agreement was shown to be effective for segmenting highly overlapping digits (Sabour et al. (2017)), but that system has several deficiencies that we have overcome in this paper:

1. It uses the length of the pose vector to represent the probability that the entity represented by a capsule is present. To keep the length less than 1, requires an unprincipled non-linearity and this prevents the existence of any sensible objective function that is minimized by the iterative routing procedure.
2. It uses the cosine of the angle between two pose vectors to measure their agreement. Unlike the negative log variance of a Gaussian cluster, the cosine saturates at 1, which makes it insensitive to the difference between a quite good agreement and a very good agreement.
3. It uses a vector of length $n$ rather than a matrix with $n$ elements to represent a pose, so its transformation matrices have $n^2$ parameters rather than just $n$.

## 8 CONCLUSION

Building on the work of Sabour et al. (2017), we have proposed a new type of capsule system in which each capsule has a logistic unit to represent the presence of an entity and a 4x4 pose matrix to represent the pose of that entity. We also introduced a new iterative routing procedure between capsule layers, based on the EM algorithm, which allows the output of each lower-level capsule to be routed to a capsule in the layer above in such a way that active capsules receive a cluster of similar pose votes. This new system achieves significantly better accuracy on the smallNORB data set than the state-of-the-art CNN, reducing the number of errors by 45%. We have also shown it to be significantly more robust to white box adversarial attacks than a baseline CNN.

SmallNORB is an ideal data-set for developing new shape-recognition models precisely because it lacks many of the additional features of images in the wild. Now that our capsules model works well on NORB, we plan to implement an efficient version to test much larger models on much larger data-sets such as ImageNet.

**ACKNOWLEDGMENTS** Thanks to Robert Gens, Eric Langlois, Taco Cohen and anonymous commentators for helpful discussions and to everyone who made TensorFlow.

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

# A APPENDIX 1: AN INTUITIVE EXPLANATION OF THE COST FUNCTION THAT IS MINIMIZED DURING DYNAMIC ROUTING

Dynamic routing is performed between two adjacent layers of capsules. We will refer to these layers as the higher-level and the lower-level. We complete the routing between one pair of layers before starting the routing between the next pair of layers. The routing process has a strong resemblance to fitting a mixture of Gaussians using EM, where the higher-level capsules play the role of the Gaussians and the means of the activated lower-level capsules for a single input image play the role of the datapoints.

We start by explaining the cost function that is minimized when using the EM procedure to fit a mixture of Gaussians. We then derive our dynamic routing procedure by making two modifications to the procedure for fitting a mixture of Gaussians.

## A.1 THE COST FUNCTION FOR FITTING A MIXTURE OF GAUSSIANS

The EM algorithm for fitting a mixture of Gaussians alternates between an E-step and an M-step. The E-step is used to determine, for each datapoint, the probability with which it is assigned to each of the Gaussians. These assignment probabilities act as weights and the M-step for each Gaussian consists of finding the mean of these weighted datapoints and the variance about that mean. If we are also fitting mixing proportions for each Gaussian, they are set to the fraction of the data assigned to the Gaussian.

The M-step holds the assignment probabilities constant and adjusts each Gaussian to maximize the sum of the weighted log probabilities that the Gaussian would generate the datapoints assigned to it. The negative log probability density of a datapoint under a Gaussian can be treated like the energy of a physical system and the M-step is minimizing the expected energy where the expectations are taken using the assignment probabilities.

The E-step adjusts the assignment probabilities for each datapoint to minimize a quantity called "free energy" which is the expected energy minus the entropy. We can minimize the expected energy by assigning each datapoint with probabilty 1 to whichever Gaussian gives it the lowest energy (*i. e.* the highest probability density). We can maximize the entropy by assigning each datapoint with equal probability to every Gaussian ignoring the energy. The best trade-off is to make the assignment probabilities be proportional to $exp(-E)$. This is known as the Boltzmann distribution in physics or the posterior distribution in statistics. Since the E-step minimizes the free energy w.r.t. the assignment distribution and the M-step leaves the entropy term unchanged and minimizes the expected energy w.r.t. the parameters of the Gaussians, the free energy is an objective function for both steps.

The softmax function computes the distribution that minimizes free energy when the logits are viewed as negative energies. So when we use a softmax in our routing procedure to recompute assignment probabilities we are minimizing a free energy. When we refit the Gaussian model of each capsule we are minimizing the same free energy *provided* the logits of the softmax are based on the same energies as are optimized when refitting the Gaussians. The energies we use are the negative log probabilities of the votes coming from a lower-level capsule under the Gaussian model of a higher-level capsule. These are not the correct energies for maximizing the log probability of the data (see the discussion of determinants below) but this does not matter for convergence so long as we use the same energies for fitting the Gaussians and for revising the assignment probabilities.

The objective function minimizes Eq. 4 which consists of:

- MDL cost $-\beta_a$ scaled by the probability of presence of capsules in layer $L + 1$ ($a_j, j \in \Omega_{L+1}$).

- Negative entropy of activations $a_j, j \in \Omega_{L+1}$.

- The expected energy minimized in M-step: sum of the weighted log probabilities ($cost_j^h$).

- Negative entropy of routing softmax assignments ($R_{ij}$) scaled by the probability of presence of the datapoint ($a_i, i \in \Omega_L$).

$$\sum_{j \in \Omega_{L+1}} a_j(-\beta_a) + a_j ln(a_j) + (1-a_j)ln(1-a_j) + \sum_h cost_j^h + \beta_u \sum_{i \in \Omega_L} r_{ij} + \sum_{i \in \Omega_L} a_i * r_{ij} * ln(r_{ij}) \quad (4)$$

## A.2 Modification 1: Mixtures of transforming Gaussians

In a standard mixture of Gaussians, each Gaussian only has a subset of the datapoints assigned to it but all of the Gaussians see the same data. If we view the capsules in the higher-layer as the Gaussians and the means of the active capsules in the lower-layer as the dataset, each Gaussian sees a dataset in which the datapoints have been transformed by transformation matrices and these matrices are different for different Gaussians. For one higher-level capsule, two transformed datapoints may be close together and for another higher-level capsule the same two datapoints may be transformed into points that are far apart. Every Gaussian has a different view of the data. This is a far more effective way to break symmetry than simply initializing the Gaussians with different means and it generally leads to much faster convergence.

If the fitting procedure is allowed to modify the transformation matrices, there is a trivial solution in which the transformation matrices all collapse to zero and the transformed data points are all identical. We avoid this problem by learning the transformation matrices discriminatively in an outer loop and we restrict the dynamic routing to modifying the means and variances of the Gaussians and the probabilities with which the datapoints are assigned to the Gaussians.

There is a more subtle version of the collapse problem that arises when different transformation matrices have different determinants. Suppose that the datapoints in a particular subset are transformed into a cluster of points in the pose space of higher-level capsule $j$ and they are transformed into a different but equally tight cluster of points in the pose space of higher-level capsule $k$. It may seem that $j$ and $k$ provide equally good models of this subset of the datapoints, but this is not correct from a generative modeling perspective. If the transformation matrices that map the datapoints into the pose space used by capsule $j$ have bigger determinants, then $j$ provides a better model. This is because the probability density of a point in the pose space of a lower-level capsule gets diluted by the determinant of the relevant transformation matrix when it is mapped to the pose of a higher-level capsule. This would be a serious issue if we wanted to learn the transformation matrices by maximizing the probability of the observed datapoints, but we are learning the transformation matrices discriminatively so it does not matter. It does, however, mean that when the dynamic routing maximizes the probability of the transformed datapoints it cannot be viewed as also maximizing the probability of the untransformed points.

The obvious way to avoid the determinant issue is to take the mean in pose space of a higher-level capsule and to map this mean back into the pose space of each lower-level capsule using the inverses of the transformation matrices. A mean in a higher-level pose space will generally map to different points in the pose spaces of different lower-level capsules because the pose of a whole will generally make different predictions for the poses of the different parts of that whole. If we use the lower-level pose space when measuring the misfit between the actual pose of a lower-level capsule and the top-down prediction of that pose obtained by applying the inverse transformation matrix to the mean of the higher-level capsule, the collapse problem disappears and we can base decisions about routing on a fair comparison of how well two different top-down predictions fit the actual pose of the lower-level capsule. We do not use this correct method for two reasons. First, it involves inverting the transformation matrices. Second, it requires a new multiplication by the inverse transformation matrices every time the higher-level mean is modified during the dynamic routing. By measuring misfits in the higher-level pose space we avoid matrix inversions and, more importantly, we avoid having to multiply by the inverses in each iteration of the dynamic routing. This allows us to do many iterations of dynamic routing for the same computational cost as one forward propagation through the transformation matrices.

## A.3 Modification 2: Mixtures of switchable transforming Gaussians

In a standard mixture of Gaussians, the modifiable parameters are the means, (co)variances, and mixing proportions and the only thing that distinguishes different Gaussians is the values of these parameters. In a mixture of transforming Gaussians, however, Gaussians also differ in the transformation matrices they use. If these transformation matrices are fixed during the fitting of the other parameters, it makes sense to have a large set of transforming Gaussians available but to only use

the small subset of them that have appropriate transformation matrices for explaining the data at hand. Fitting to a dataset will then involve deciding which of the transforming Gaussians should be "switched on". We therefore give each transforming Gaussian an additional activation parameter which is its probability of being switched on for the current dataset. The activation parameters are not mixing proportions because they do not sum to 1.

To set the activation probability for a particular higher-level capsule, $j$, we compare the description lengths of two different ways of coding the poses of the activated lower-level capsules assigned to $j$ by the routing, as described in section 3. "Description length" is just another term for energy. The difference in the two description lengths (in nats) is put through a logistic function to determine the activation probability of capsule $j$. The logistic function computes the distribution $(p, 1 - p)$ that minimizes free energy when the difference in the energies of the two alternatives is the argument of the logistic function. The energies we use for determining the activation probabilities are the same energies as we use for fitting the Gaussians and computing the assignment probabilities. So all three steps minimize the same free energy but with respect to different parameters for each step.

In some of the explanations above we have implicitly assumed that the lower-level capsules have activities of 1 or 0 and the assignment probabilities computed during the dynamic routing are also 1 or 0. In fact, these numbers are both probabilities and we use the product of these two probabilities as a multiplier on both the baseline description length of each lower-level mean and its alternative description length obtained by making use of the Gaussian fitted by a higher-level capsule.

## B  SUPPLEMENTARY FIGURES

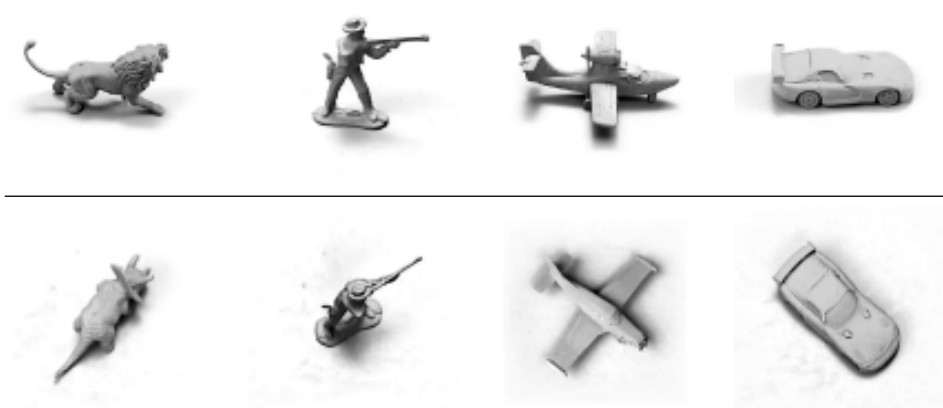

Figure B.1: Sample smallNORB images at different viewpoints. All images in first row are at azimuth 0 and elevation 0. The second row shows a set of images at a higher-elevation and different azimuth.

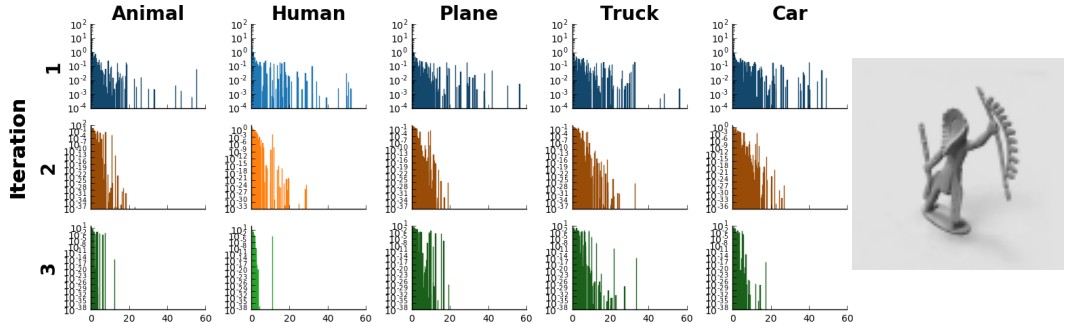

Figure B.2: Log scale histogram of distances between the receiving votes and the center of each of the 5 final capsules. The three rows show the 5 histograms for iterations 1, 2 and 3. Unlike Fig. 2 the histograms are independently log scaled so that small and large counts can both be seen. Also, the considered distance range is 60 and the number of bins is much larger.

Figure B.3: Adverserial images generated with FGSM with $\epsilon = 0.1$ and $\epsilon = 0.4$ on the CNN model and the Capsule model.

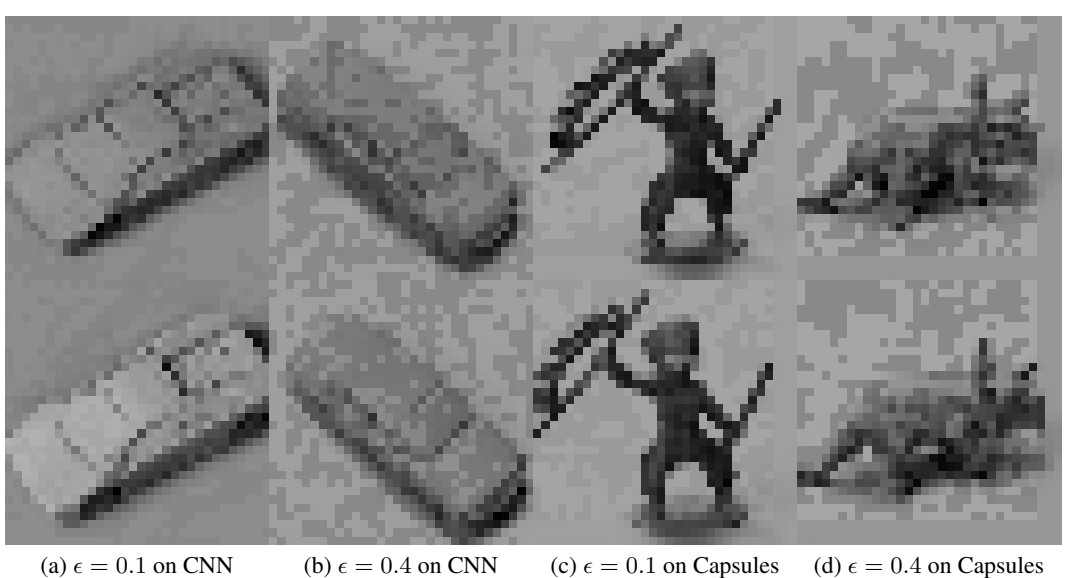

(a) $\epsilon = 0.1$ on CNN     (b) $\epsilon = 0.4$ on CNN     (c) $\epsilon = 0.1$ on Capsules     (d) $\epsilon = 0.4$ on Capsules

