# OpenReview forum: "Matrix capsules with EM routing"
_ICLR.cc/2018/Conference — Accept (Poster)_

### Official Review · AnonReviewer1 · 2017-11-27
**A novel approach for capsule networks**

**Rating:** 7
**Confidence:** 3

**Review:**

The paper proposes a novel architecture for capsule networks. Each capsule has a logistic unit representing the presence of an entity plus a 4x4 pose matrix representing the entity/viewer relationship. This new representation comes with a novel iterative routing scheme, based on the EM algorithm.
Evaluated on the SmallNORB dataset, the approach proves to be more accurate than previous work (beating also the recently proposed "routing-by-agreement" approach for capsule networks by Sabour et al.). It also generalizes well to new, unseen viewpoints and proves to be more robust to adversarial examples than traditional CNNs.

Capsule networks have recently gained attention from the community. The paper addresses important shortcomings exhibited by previous work (Sabour et al.), introducing a series of valuable technical novelties.
There are, however, some weaknesses. The proposed routing scheme is quite complex (involving an EM-based step at each layer); it's not fully clear how efficiently it can be performed / how scalable it is. Evaluation is performed on a small dataset for shape recognition; as noted in Sec. 6, the approach will need to be tested on larger, more challenging datasets. Clarity could be improved in some parts of the paper (e.g.: Sec. 1.1 may not be fully clear if the reader is not already familiar with (Sabour et al., 2017); the authors could give a better intuition about what is kept and what is discarded, and why, from that approach. Sec. 2: the sentence "this is incorrect because the transformation matrix..." could be elaborated more. V_{ih} in eq. 1 is defined only a few lines below; perhaps, defining the variables before the equations could improve clarity. Sec. 2.1 could be accompanied by mathematical formulation).
All in all, the paper brings an original contribution and will encourage further research / discussion on an important research question (how to effectively leverage knowledge about the part-whole relationships).

Other notes:
- There are a few typos (e.g. Sec. 1.2 "(Jaderberg et al. (2015)",  Sec. 2 "the the transformation", Sec. 4 "cetral crop" etc.).
- The authors could discuss in more detail why the approach does not show significant improvement on NORB with respect to the state of the art.
- The authors could provide more insights about why capsule gradients are smaller than CNN ones.
- It would be interesting to discuss how the network could potentially be adapted, in the future, to: 1. be more efficient 2. take into account other changes produced by viewpoint changes (pixel intensities, as noted in Sec. 1).
- In Sec, 4, the authors could provide more details about the network training.
- In Procedure 1, for indexing tensors and matrices it might be better to use a comma to separate dimensions (e.g. V_{:,c,:} instead of V_{:c:}).

---

> ### Author Response · Authors · 2018-01-08
> **re: A novel approach for capsule networks**
>
> Thank you for your detailed reading of the paper and suggestions!
> As per your comments on the EM routing, we agree that it was not presented as best it could have been, and have added an appendix to present a gentle and thorough introduction to the free energy view of EM and the objective function which our routing operation minimizes. In response to the question about efficiency, we would like to draw your attention to the total number of arithmetic operations required for the routing procedure - each iteration of routing represents fewer arithmetic operations than a single layer feed forward pass, but due to architectural optimization decisions in tensorflow, our current capsule implementation is not as fast as it could be.
>
> We agree that larger scale testing would ideal, but due to the aforementioned efficiency limitations were not able to include it in this paper.
>
> In regards to your other comments we have done the following:
> - To increase the clarity of the paper,  we have made several changes to the language used, and improved the mathematical notation.
> - We have added section 2 which provides an intuitive explanation of capsules and makes clear when the routing occurs. We feel that improves the readers' ability to engage with the rest of the presented content. We also defined the variables and notation used in the rest of the paper more explicitly.
> - We have expanded on the sentence "this is incorrect because the transformation matrix..." you mentioned which is now in the appendix.
> - We have also made several changes to the nation and language throughout the paper to make it more comprehensible.
> thank you for your feedback, and hope that we have addressed your comments to your satisfaction.

---

### Official Review · AnonReviewer3 · 2017-11-27
**Idea is interesting; need more empirical validation than smallNORB**

**Rating:** 6
**Confidence:** 3

**Review:**

This paper proposes a new kind of capsules for CNN. The capsule contains a 4x4 pose matrix motivated by 3D geometric transformations describing the relationship between the viewer and the object (parts). An EM-type of algorithm is used to compute the routing.

The authors use the smallNORB dataset as an example. Since the scenes are simulated from different viewer angles, the pose matrix quite fits the motivation. It would be more beneficial to know if this kind of capsules is limited to the motivation or is general. For example, the authors may consider reporting the results of the affNIST dataset where the digits undergo 2D affine transformations (in which case perhaps 3x3 pose matrices are enough?).

Minor: The arguments in line 5 of the procedure RM Routing(a,V) do not match those in line 1 of the procedure E-Step.

Section 2.1 (objective of EM) is unclear. The authors may want to explicitly write down the free energy function.

The section about robustness against adversarial attacks is interesting.

Overall the idea appears to be useful but needs more empirical validation (affNIST, ImageNet, etc).

---

> ### Author Response · Authors · 2018-01-08
> **affNIST generalization and EM objective**
>
> thank you for the feedback! To address your comments we have done the following:
> - To clarify the EM objective we have added an extended and thorough appendix which presents a gentle and intuitive explanation of the free energy view of EM, and explicit free energy function, and how our routing algorithm makes use of it.
> - We believe that the benefit of capsules is not limited to smallNORB and will generalize. As suggested, we replicated the affNIST generalization experiment reported in the previous Capsule paper (Sabour et al. 2017). We found that our EM capsule model (the exact architecture used for smallNORB and MNIST in the paper), when trained to 0.8% test error on expanded MNIST (40x40 pixel MNIST images, created by padding and shifting MNIST), achieved 6.9% test error on affNIST. We trained a baseline CNN (with AlexNet architecture, without pooling) to 0.8% test error and it was only able to achieve 14.1% test error on affNIST. Our capsule model was able to half the test error of a CNN when trained on MNIST and tested on affNIST.  Due to time and space constraints these results are not reported in the paper as it is now.
> - finally we address the minor issue raised in line 5 of the routing procedure.
> we hope this has addressed your concerns, and thank you for your suggestions.

---

### Official Review · AnonReviewer2 · 2017-12-01
**An extremely opaque paper with a potentially interesting idea and good results**

**Rating:** 4
**Confidence:** 2

**Review:**

The paper describes another instantiation of "capsules" which attempt to learn part-whole relationships and the geometric pose transformations between them.  Results are presented on the smallNORB test set obtaining impressive performance.

Although I like very much this overall approach, this particular paper is so opaquely written that it is difficult to understand exactly what was done and how the network works.  It sounds like the main innovation here is using a 4x4 matrix for the pose parameters, and an iterative EM algorithm to find the correspondence between capsules (routing by agreement).  But what exactly the pose matrix represents, and how they get transformed from one layer to the next, is left almost entirely to the reader's imagination.  In addition, how EM factors in, what the probabilities P_ih represent, etc. is not clear.  I think the authors could do a much better job explaining this model, the rationale behind it, and how it works.

Perhaps the most interesting and compelling result is Figure 2, which shows how ambiguity in object class assignment is resolved with each iteration.  This is very intriguing, but it would be great to understand what is going on and how this is happening.

Although the results are impressive, if one can't understand how this was achieved it is hard to know what to make of it.

---

> ### Author Response · Authors · 2018-01-08
> **Improvements on the clarity of the paper**
>
> Thank you for your comments. upon reflection we agree that the paper was confusing and we have taken several steps to reduce the opacity of our work to the reader. To that end we have done the following:
> - We have added section 2 which gives a general and intuitive explanation of the mechanism of capsule networks, paying close attention to how pose matrices get transformed from one layer to the next.
> - Having identified the EM objective as another source of confusion, we added an extended appendix in which we provide a gentle and approachable explanation for the free energy view of EM and how our routing algorithm builds upon it.
> - We have also added a paragraph to further explain figure 2 in the experiments section.
> - Finally we have made several changes to the language of the paper, focusing in particular on the notation.
> We believe that the comprehensibility of the paper has thus improved and appreciate your criticism.

---

### Public Comment · (anonymous) · 2017-11-01
**state-of-the-art on "small NORB"**

1.5% error rate has previously been reported on small NORB.
https://www.researchgate.net/publication/265335724_Nonlinear_Supervised_Locality_Preserving_Projections_for_Visual_Pattern_Discrimination

---

> ### Author Response · Authors · 2017-11-01
> **state-of-the-art on "small NORB"**
>
> They gain a lot by using the meta data at test time. Without using that information (which normally is not available at test time) they get 2.6%.

---

> > ### Public Comment · (anonymous) · 2017-11-02
> > **state-of-the-art on "small NORB"**
> >
> > The meta data is not used during test time only during training time.

---

### Public Comment · ~Jianfei_Chen1 · 2017-11-11
**The objective function**

Can you write down what exactly is the objective function in Section 2.1?

---

> ### Author Response · Authors · 2017-11-25
> **Re: The objective function**
>
> The objective function in details is:
> \sum_c a'_c (-\beta_a) + a'_c ln(a'_c) + (1-a'_c)ln(1-a'_c)+\sum_h cost_{ch} + \sum_i a_i *  r_{ic} * ln(r_{ic})
>
> a'_c is the activation for capsule c in layer L+1 and a_i is the activation probability for capsule i in layer L. The rest of the notations follow paper.
>
> Plots showing the decay of objective function and the absolute difference between two routing iterations in the above objective function can be found at:
> https://imgur.com/a/eeD2X

---

### Public Comment · (anonymous) · 2017-11-11
**How to transform conv layer to the primary capsule layer?**

I still don't understand the transformation from convolution layer to primary capsule layer? Is it achieved by slicing 4x4*32 patches from the conv layer and then do a linear transformation for each 4x4 matrices?  what is the weight in "The activations of the primary capsules are produced by applying the sigmoid function to weighted sums of the same set of lower layer ReLUs." is it the 4x4 variable? I found it confusing, can you elaborate how this works.

---

> ### Public Comment · ~Jianfei_Chen1 · 2017-11-12
> **How to transform conv layer to the primary capsule layer?**
>
> Figure 1 explains that. I guess they use a A*B*(4*4+1) kernel to (linear) transform a 1 width * 1 height * 32 channels patch to 32 capsules, each shape is 4*4+1. Then they reshape the 4*4 part as a matrix and apply a sigmoid on the 1 part.

---

> ### Author Response · Authors · 2017-11-25
> **How to transform conv layer to the primary capsule layer?**
>
> As Jianfei has explained, the primary capsule layer is a convolutional layer with 1x1 kernel. It transforms the A channels in the first layer to B*(4x4+1) channels. Then we split the B*(4x4+1) channels into B*(4x4) as the pose matrices for B capsules and B*1 as the activation logits of B capsules in primary layer. Then we apply sigmoid nonlinearity on the activation logits.

---

### Public Comment · ~Gavin_Weiguang_Ding1 · 2017-11-14
**dimensionality of transformation matrix W_{ic} in ConvCaps**

In the convolutional capsule layers, what's the dimensionality of  transformation matrix W_{ic}?
Is it still (4*4)->(4*4) which correspond to a 1*1 linear convolutional layer?
or it is (4*4*k*k)->(4*4) which correspond to a k*k linear convolutional layer?

---

> ### Author Response · Authors · 2017-11-25
> **dimensionality of transformation matrix W_{ic} in ConvCaps**
>
> W_{ic} is 4*4 if you flatten the capsule types and grid positions. Therefore i goes over changes in the range of (1, channels * height * width) in this formulation.
>
> However, We share the W_ic between different positions of two capsule types as in a convolutional layer with a kernel size k. Therefore, the total number of trainable parameters between two convolutional capsule layer types is 4*4*k*k and for the whole layer is 4*4*k*k*B*C. Where B is the number of different capsule types in layer bellow and C is the number of different capsule types in the next layer.
>
> Please note that it is 4*4 rather than (4*4)*(4*4).

---

### Public Comment · ~Micha_Pfeiffer1 · 2017-11-29
**V_ih**

1) "V_ih is the product of the the transformation matrix W_ic that is learned discriminatively"
There is part of the sentense missing. Also, I believe this sentence describes "V_i" and not "V_ih". Suggestion:
" ... and V_ih is the value on dimension h of the vote V_i from capsule i to capsule c. V_i is obtained by taking the matrix product of the pose p_i of capsule i and the transformation Matrix W_ic. W_ic is learned discriminatively."

2) From what I understand, the vote V_i is a matrix (since it's obtained by multiplying a 4x4 matrix with a 4x4 matrix), and v_ih is a scalar. I found "V_ih is the value on dimension h of the vote ..." to be missleading. Maybe it should be mentioned that V_i has to be reshaped into a vector first and then its h'th entry is V_ih?

---

### Public Comment · ~Kaitlin_Duck_Sherwood1 · 2017-11-29
**Typo**

The sentence fragment:
   Spatial transformer networks (Jaderberg et al. (2015) seeks
is missing a ), and the subject is plural and not singular.  So it should be:
    Spatial transformer networks (Jaderberg et al. (2015)) seek

---

### Public Comment · (anonymous) · 2017-12-01
**Spread loss is squared WW-Hinge-Loss**

The spread-loss in 3.1 is the square of the WW-hinge-loss for multi-class SVMs, a large-margin loss.

See:

Weston, Jason; Watkins, Chris (1999). "Support Vector Machines for Multi-Class Pattern Recognition" (PDF). European Symposium on Artificial Neural Networks.

and the following paper describes the relations of the different variants of this loss:

http://jmlr.org/papers/v17/11-229.html
In the notation of that paper, it would be the combination of sum-over-others aggregation with relative margin concept and squared hinge loss.

For theoretical considerations, the log-probability should be used, in which case m  = 1 is fine and the last layer would not need to be normalized any more.

---

### Public Comment · (anonymous) · 2017-12-08
**beta_v and beta_a**

The dimensionality of the two trained beta parameters is not very clear to me from the paper. Are they shared across all capsules in the same layer (making them scalars) or does each capsule type have their own beta (meaning they are vectors).  I have had a look at the current implementation attempts of the model on GitHub and there the interpretations vary widely as well. Could you please clarify this point?

Minor notes on the algorithm (Procedure 1):
- "V_ich  is an H dimensional vote...": Did you mean V_ic?
- M-Step line 5: Missing quantifier. Like mu and sigma, cost_h is computed for all h

---

> ### Author Response · Authors · 2017-12-08
> **re: beta_v and beta_a**
>
> beta_v and beta_a are per capsule type. Therefore, they are vectors for both convolutional capsules and final capsules. For example in terms of the notation in fig.1 beta_a and beta_v for convCaps1 are C dimensional vectors.
>
> Thanks! We will revise the paper in regard to these points.

---

### Public Comment · ~Arent_Warren_de_Jong1 · 2017-12-11
**Spatial downsampling from ConvCaps2 (L_final-1) to Class Capsules (L_final)**

Thanks for all your research effort. It is great to read on this new paradigm and to see it actually working.

One part that is missing in my opinion, or I am very ignorantly glossing over it, is the downsampling from ConvCaps2 (L_final-1) to Class Capsules (L_final).

As mentioned, weights are shared among same entity capsules, so this would result in a one dimensional convolution (because K=1, stride=1), i.e. keeping the two spatial dimensions of the ConvCaps2 layer.
Whereas the other layer transitions indeed keep their spatial information and result in multiple, same-entity capsules spread over the 2 input image dimensions, the final layer only has one capsule for each class for the entire image.
IMHO this therefore requires a downsampling of the ConvCaps2 votes, a la maxpool, averagepool, or some extra dimension added to the EM routing algorithm.

---

### Public Comment · ~Hang_Yu2 · 2017-12-23
**Matrix Capsule With EM Routing Reproduce Report**

Author: Hang Yu | Suofei Zhang

Email: hangyu5 at illinois.edu | zhangsuofei at njupt.edu.cn

## Reproduce Method

#### Hyperparameters
smallNORB dataset:
* Samples per epoch: 46800
* Sample dimensions: 96x96x1
* Batch size: 50
* Preprocessing:
    * training:
        1. add random brightness with max delta equals 32 / 255.
        2. add random contrast with lower 0.5 and upper 1.5.
        3. resize into HxW 48x48 with bilinear method.
        4. crop into random HxW 32x32 piece.
        5. apply batch norm to have zero mean and unit variance.
        6. squash the image from 4 so that each entry has value from 0 to 1. This image is to be compared with the reconstructed image.
    * testing:
        1. resize into HxW 48x48 with bilinear method.
        2. crop the center HxW 32x32 piece.
        3. apply batch norm with moving mean and moving variance collected from training data set.

#### Method

1. The so called dynamic routing is in analog to the fully-connected layer in CNN. The so called ConvCaps structure extends dynamic routing into convolutional filter structure. The ConvCaps are implemented similarly as the dynamic routing for the whole feature map. The only difference is to tile the feature map into kernel-wise data and treat different kernels as batches. Then EM routing can be implemented within each batch in the same way as dynamic routing.

2. Different initialization strategies are used for convolutional filters. Linear weights are initialized with Xavier method. Biases are initialized with truncated normal distribution. This configuration provide higher numerical stability of input to EM algorithm.

3. The output of ConvCaps2 layer is processed by em routing with kernel size of 1*1. Then a global average pooling is deployed here to results final Class Capsules. Coordinate Addition is also injected during this stage.

4. Equation 2 in E-step of Procedure 1 from original paper is replaced by products of probabilities directly. All the probabilities are normalized into [0, 10] for higher numerical stability in products. Due to the division in Equation 3, this operation will not impact the final result. Exponent and logarithm are also used here for the same purpose.

5. A common l2 regularization of network parameters is considered in the loss function. Beside this, reconstruction loss and spread loss are implemented as the description in the original paper.

6. Learning rate: starts from 1e-3, then decays exponentially in a rate of 0.8 for every 46800/50 steps, and ends in 1e-5 (applied for all trainings).

7. We use Tensorflow 1.4 API and python programming language.

## Reproduce Result

#### Overview

Experiments on is done by Suofei Zhang. His hardware is:

* cpu：Intel(R) Xeon(R) CPU E5-2680 v4@ 2.40GHz，
* gpu：Tesla P40


**On test accuracy**:

smallNORB dataset test accuracy (our result/proposed result):

* CNN baseline (4.2M): 88.7%(best)/94.8%
* Matrix Cap with EM routing (310K, 2 iteration): 91.8%(best)/98.6%

There are two comments to make:

1. Even though the best of Matrix Cap is over by 3% to the best of CNN baseline, the test curve suggest Matrix Cap fluctuates between roughly 80% to 90% test dataset.
2. We are curious to know the learning curve and test curve that can be generated by the author.

**Training speed**:

1. CNN baseline costs 6m to train 50 epochs on smallNORB dataset. Each batch costs about 0.006s.
2. Matrix Cap costs 15h55m36s to train. Each batch costs about 1.2s.

**Recon image**:

Will come soon.

**routing histogram**:

We have difficulty in understanding how the histogram is calculated.

**AD attack**:

We haven't planned to run AD attack yet.

### Notes

> **Status:**
> According to github commit history, this reproduce project had its init commit on Nov.19th. We started writing this report on Dec.19th. Mainly, it is cost by undedicated code review so that we have to fix bug and run it again, otherwise the project should be able to finish in a week.

> **Current Results on smallNORB:**
- Configuration: A=32, B=8, C=16, D=16, batch_size=50, iteration number of EM routing: 2, with Coordinate Addition, spread loss, batch normalization
- Training loss. Variation of loss is suppressed by batch normalization. However, there still exists a gap between our best results and the reported results in the original paper.

- Test accuracy(current best result is 91.8%)

> **Current Results on MNIST:**
- Configuration: A=32, B=8, C=16, D=16, batch_size=50, iteration number of EM routing: 2, with Coordinate Addition, spread loss, batch normalization, reconstruction loss.

- Test accuracy(current best result is 99.3%, only 10% samples are used in test)

##Reference

[1] [MATRIX CAPSULES WITH EM ROUTING (paper)](https://openreview.net/pdf?id=HJWLfGWRb)

[2] [Matrix-Capsules-EM-Tensorflow (our github repo: code and comments)](https://github.com/www0wwwjs1/Matrix-Capsules-EM-Tensorflow/)

---

### Public Comment · ~Guido_Sales_Calvano1 · 2018-07-18
**How are the lambda and margin parameters changed during training?**

There are still a few gaps in the paper concerning the used lambda parameter during em routing and the margin parameter m for spread loss;

Lambda:
"the inverse temperature λ increases at each iteration with a fixed schedule"
What is meant by an iteration? The context implies that you mean lambda is assigned an initial value every time em routing is executed, and that lambda is then increased each em routing iteration, but a different interpretation is that it increases for each training iteration, i.e. as a function of the number of examples fed to the network?
What is the initial value, and how is the value changed during training, i.e. what is the "fixed schedule" the paper talks about? Does the parameter increase exponentially or linearly?

Margin parameter m:
"By starting with a small margin of 0.2 and linearly increasing it during training to 0.9, we avoid dead capsules in the earlier layers."

To reproduce this linear increase it is necessary to know how over how many epochs this linear increase takes place. How many epochs of training did you have to do?

---

> ### Author Response · Authors · 2018-07-31
> **Lambda and margin**
>
> the formula we used for lambda is:
> lambda = final_lambda * (1 - tf.pow(0.95, tf.cast(i + 1, tf.float32)))
> where 'i' is the routing iteration (range is 0-2). Final_lambda is set to 0.01.
>
> The margin that we set is:
> margin = 0.2 + .79 * tf.sigmoid(tf.minimum(10.0, step / 50000.0 - 4))
> where step is the training step. We trained with batch size of 64.

---

### Public Comment · ~Guido_Sales_Calvano1 · 2018-07-31
**Some clarification on the convolution topology?**

The paper states that capsules are connected through convolutions. However, these can be interpreted in two ways:

1. An image (of features or color channels) is broken up into patches that are entirely isolated from each other. For each kernel for each position in the output an input capsule is assigned to an output capsule separate from assignments in other kernels. I.e. output capsules at different positions do not compete for input capsules.

2. However, another interpretation is  the e-step takes every output capsule into account in whose kernel the input capsule appears. I.e. output capsules at different positions do compete for input capsules.

Which interpretation will reproduce your results?

---

> ### Author Response · Authors · 2018-07-31
> **convolution capsule layer**
>
> The first option is like having 1x1 convolution layers. The second option is what happens if you have kernel size larger than one. Since we have 3x3 convolution capsule layers (32 capsule types each) it means that 9x32 capsules receive the vote of a single capsule in layer bellow. Therefore these 9x32 capsules are competing for its vote (normalize the routing factors over the feedback of these 3x3x32 capsules).

---

### Public Comment · (anonymous) · 2018-10-26
**Regularization and learning rate?**

Did you use any regularization in this paper? Perhaps a decoder network as in Sabour et al. (2017), or weight decay of some sort?

Also, what learning rate and schedule did you use?

---

> ### Author Response · Authors · 2018-10-29
> **regularizer & learning rate**
>
> We use a weight decay loss with a small factor of .0000002 rather than the reconstruction loss.
> We use an exponential decay with learning rate: 3e-3, decay_steps: 20000, decay rate: 0.96.

---

> > ### Public Comment · (anonymous) · 2018-10-30
> > **Where to apply regularization?**
> >
> > Thanks, that's really helpful.
> >
> > Where do you apply the regularization that you mentioned?
> > - initial 5x5 convolution (ReLU Conv1)
> > - 1x1 linear transformation (PrimaryCaps)
> > - pose parameter convolution weights (ConvCaps1, ConvCaps2, ClassCaps)
> > - beta_v and beta_a

---

### Public Comment · ~Guido_Sales_Calvano1 · 2018-10-28
**More parameters**

And to add to the question below;

* What optimizer did you use (and what parameters)
* How did you initialize your weights and did you take any measures to deal with gradient issues for your sigmoid curves?
* Did you do any special ordering of your training batches?
* Do you fill your batches in any special way?

---

> ### Public Comment · ~Guido_Sales_Calvano1 · 2018-10-28
> **The correct input count of xavier initialization?**
>
> If you were to (or indeed did) use xavier initialization to prevent output and gradient issues with your sigmoid curves, did/would you use for the input count
> 1.  4, perceiving each weight matrix as a small dense 4 neuron subnetwork (conveniently creating weight vectors of with a mean length close to 1) or
> 2. the kernel size for the input
> 3. the kernel size * 4 thus combining both perspectives
>
> What is/would be your output neuron count?

---

> > ### Author Response · Authors · 2018-10-29
> > **Optimizer & initializer**
> >
> > We use Adam optimizer with default Tensorflow parameters.
> >
> > We did not have any special ordering of training batches and we random shuffle. In terms of TF batch:
> > capacity=2000 + 3 * batch_size,
> > # Ensures a minimum amount of shuffling of examples.
> > min_after_dequeue=2000.
> >
> > Please not that for smallnorb viewpoint generalization test to make sure that the model only sees a fraction of certain directions and the generalization test is strict, we calibrated the semantic of azimuth '0' for every class so that the object at azimuth '0' roughly heads to the right. Therefore, in training the model never sees an object which heads to the left (from any class) while in test it is tested on objects which head to the left as well.
> >
> > The gradient indeed is tricky. Almost all the math is done in log-scale to avoid numerical issues. We used truncated_normal_initializer and set the std so that at the start of training half of the capsules in each layer are active and half inactive (for the Primary Capsule layer where the activation is not computed through routing we use different std for activation convolution weights & for pose parameter convolution weights).
> >
> > Recently we are using a new initialization method: every 4x4 is initialized with I + noise of 0.03: (1 on the diag, random uniform noise in the range +/- 0.03 everywhere else). This new method is more scale able and easier to train.

---

> > > ### Public Comment · ~Guido_Sales_Calvano1 · 2018-11-06
> > > **Could you elaborate in the initialization of the poses of the primary capsules**
> > >
> > > My apologies if I ask too many details but please understand that I only have one GPU at my disposal through my work (i.e. one run costs 8 days), and can only spend spare time (i.e. weekends/evenings) on this. So every parameter I have to figure out experimentally will cost me an incredible amount of scarce time.
> > >
> > > Initialization of the poses of the primary capsules still does not seem trivial to me, because stabilizing input/output variance is arguably not the best approach:
> > >
> > > For the chains of 4x4 matrices formed by em routing to not have exploding/vanishing output/gradients it seems like the determinant and eigenvectors matter more than the variance, i.e. not inflating or deflating poses. This would also explain your I + .03 * noise approach. Therefore I expect you might have used a different weight initialization for the primary caps poses than xavier. If you could be a bit more specific on how you determined those weights as well I would appreciate it a lot. I can assume the activations of the primary capsules will be just fine using xavier initialization right?
> > >
> > > Finally if you could share any of the standard deviations you used in your initial approach, I would appreciate it very much as well.

---

### Public Comment · ~Guido_Sales_Calvano1 · 2018-10-28
**Hardware/execution time**

What hardware did you use to train the network, and how long did it take until training was completed?

---

> ### Author Response · Authors · 2018-10-29
> **Hardware**
>
> We used 8 sync gpus (batch of 64, 8 on each gpu) to train for ~ a day on small norb, ~10hr on MNIST and ~ 2 day on Cifar10.

---

### Public Comment · (anonymous) · 2018-10-30
**Backpropagation through EM**

Did you use tf.stop_gradient so that the gradient does not flow through each iteration of the EM routing, only the last step? Or did you allow the gradient to flow through all iterations of EM routing?

---

> ### Author Response · Authors · 2018-10-30
> **BP through EM**
>
> The gradient flows through EM algorithm. We do not use stop gradient. A routing of 3 is like a 3 layer network where the weights of layers are shared.

---

### Public Comment · (anonymous) · 2018-11-01
**Where to apply regularization?**

Where do you apply the regularization that you mentioned?
- initial 5x5 convolution (ReLU Conv1)
- 1x1 linear transformation (PrimaryCaps)
- pose parameter convolution weights (ConvCaps1, ConvCaps2, ClassCaps)
- beta_v and beta_a

---

### Public Comment · ~Ashley_Gritzman1 · 2019-08-23
**Open Source Implementation**

Here is our implementation where we get a closer to the accuracy reported in the paper. We get 95.4% on smallNORB, whereas the paper reports an accuracy of 97.8% (configuration: A=64, B=8, C=D=16).

https://github.com/IBM/matrix-capsules-with-em-routing

---

> ### Author Response · Authors · 2019-08-23
> **Original implementation**
>
> Thank you for the implementation and enlightening the challenges. We are looking into it.
>
> We open sourced our code on January here:
> https://github.com/google-research/google-research/commits/master/capsule_em
> which provides the checkpoints  for 1.3% test error too.

---

### Public Comment · ~Henna_John1 · 2020-08-22
**Does NutraVesta Proven Plus Immune Boost Formula Work?**

Only a stronger immune system can help keep you fit, keeping you from getting sick again and again. If you think your immunity needs a boost, Nutravesta Proven plus is a very affordable formula that will work on your immune system, making it potent enough to fight disease and save you from paying medical bills. costly every month. With this product at your fingertips, you can happily enjoy foods that otherwise seemed to make you sick! See more https://www.phdsc.org/NutraVesta-Proven-Plus-Review

---

### Public Comment · ~Eugene_Blaze1 · 2020-08-27
**Meticore Review – Is It A Genuine Weight Loss Supplement?**

According to Meticore Review, Meticore is a weight loss supplement and is famous for its uniqueness. The main reason behind this is, Meticore offers you six best plants and nutrients that are perfect for weight loss. One of the fuels for weight loss is low core temperature. When the body maintains a low core temperature, it is straightforward to supercharge metabolism. This secret can be applied, and it gives the best results both in men and women. Visit website https://rubinstein-taybi.org/meticore-review/

---

### Public Comment · ~liam_alexander1 · 2020-08-29
**Man Greens Review- Does This Powder Help To Boost Man’s Testosterone Level?**

According to Man Greens Review, Man greens powder is clinically proved supplement that is a mixture of Superfoods and ingredients best known for potency and it is clinically tested as effective. The ingredients present in Man Greens supplement were used 6000 years ago by ancient Ayurveda, and are used in the current world that will help in boosting a man’s testosterone level without provoking other hormonal balance. Visit website https://systemagility.com/man-greens-review/

---

### Public Comment · ~Deyproject_blog1 · 2020-09-07
**Reading Head Start Review: Teach And Improve Your Child’s Reading Skills**

Reading Head Start program is a learning program aimed at teaching and improving the reading skills of your child. The teaching techniques involved in this program are scientifically proven and she has full confidence in the ability of the program that she offers a 100%, 365-day money-back guarantee.Reading Head Start Review says that it tries to make the content as fun and educational as possible because the intent is not only to help your child read but also to get them interested in books.
Visit website https://www.deyproject.org/reading-head-start-review/

---

### Decision · Program_Chairs · 2018-01-29
**ICLR 2018 Conference Acceptance Decision**

**Decision:**

Accept (Poster)

**Comment:**

Authors present a new multi-layered capsule network architecture, implemented an EM routing procedure, and introduced "Coordinate Addition".  Capsule architectures are gaining interest because of their ability to achieve equivariance of parts, and employ a new form of pooling called "routing" (as opposed to max pooling) which groups parts that make similar predictions of the whole to which they belong, rather than relying on spatial co-locality. New state-of-art performances are being achieved on focused datasets, for which the authors have continued the trend.

Pros:
- New significant improvement to state-of-art performance is obtained on smallNORB, both in comparison to CNN structure as well as the most recent previous implementation of capsule network.

Cons:
- Some concern arose regarding the writing of the paper and the ability to understand the material, which authors have made an effort to address.

Given the general consensus of the reviewers that this work should be accepted, the general applicability of the technology to multiple domains, and the potential impact that improvements to capsule networks may have on an early field, area chair recommends this work be accepted as a poster presentation.